# New Kinetic Investigations to Better Understand the Mechanism of Polymorphic Transformations of Pharmaceutical Materials Induced by Milling

**DOI:** 10.3390/pharmaceutics17111404

**Published:** 2025-10-30

**Authors:** Mathieu Guerain, Anthony Dupont, Florence Danède, Darina Barkhatova, Jean-François Willart

**Affiliations:** UMR 8207-UMET-Unité Matériaux et Transformations, Centre National de la Recherche Scientifique, Institut National de Recherches Agronomiques, Ecole Nationale Supérieure de Chimie de Lille, Université de Lille, F-59650 Villeneuve d’Ascq, France; mathieu.guerain@univ-lille.fr (M.G.); anthony.dupont@univ-lille.fr (A.D.); florence.danede@univ-lille.fr (F.D.); darina.barkhatova@ist.ac.at (D.B.)

**Keywords:** pharmaceuticals, polymorphic transformations, milling, molecular materials, amorphous state, differential scanning calorimetry, X-ray diffraction, synchrotron facility

## Abstract

**Objective:** The aim of this work is to improve the understanding of the mechanisms underlying the polymorphic transformations of pharmaceutical materials during milling. Elucidating these mechanisms is essential for controlling the polymorphism of active pharmaceutical ingredients and thereby improving their performance. **Method**: The structural evolution of various pharmaceutical compounds (sulfamerazine, glycine, mannitol, and famotidine) upon milling was followed using ex situ laboratory X-ray diffraction and in situ synchrotron measurements, complemented by DSC analyses. **Results**: For each compound, the kinetics of the polymorphic transformation was found to be sigmoidal and the presence of an intermediate amorphous phase during the transition from the initial to the final polymorphic form was also identified. **Conclusions**: The kinetic data obtained for sulfamerazine and glycine, together with the detection of an amorphous intermediate during the transformations of mannitol and famotidine, support the conclusion that milling-induced polymorphic transformations in pharmaceutical materials generally proceed via an amorphization–recrystallization mechanism.

## 1. Introduction

During the formulation of a drug, from the initial powder to the final tablet, the various processing steps can induce phase transformations of the active pharmaceutical ingredients (API) [1,2,3,4]. In particular, milling—commonly used to reduce the particle size of APIs—is highly susceptible to inducing such transformations. When performed at high energy, it can lead to an amorphization of the material if the milling temperature is below the glass transition temperature (Tg) of the material. It can also lead to polymorphic transformation when the milling temperature is above the Tg of the material [5,6,7]. In this case, the change in the crystallographic configuration of the API must be controlled, because its polymorphic form can influence both its physical stability and bioavailability. However, the mechanisms that govern these polymorphic transformations remain poorly understood. In particular, the exact pathway of transformations is still unclear. It has been proposed that this transformation is achieved either through a direct displacement of molecules in the crystal lattice, or via a transient amorphization of the initial form followed by an extremely rapid recrystallization toward the final form [8].

A literature review was recently performed to identify and list polymorphic transformations of pharmaceutical materials induced by milling [9], and in view of the results, to propose a mechanism capable of describing all milling-induced polymorphic transformations. However, certain information was found to be lacking to fully validate the proposed mechanism, and additional experiments were necessary to confirm it. These experimental works and their results are detailed here.

The mechanism of polymorphic transformation emerging from the literature review is based on an amorphization–recrystallization mechanism that leads from the initial polymorphic form to the final polymorphic form. This mechanism exhibits the following key features:

i. Polymorphic transformations do not occur directly, but instead proceed through a transient amorphous stage of the initial polymorphic form. In this process, mechanical shocks locally induce the amorphization of the material, followed by recrystallization into the new polymorph. The observation of the transient amorphous matter is strongly linked to its recrystallization rate.

i.a. When the glass transition temperature (Tg) is close to or slightly higher than the milling temperature (Tmill), the amorphous phase has low molecular mobility and thus a noticeable lifetime so that it can be observed during the polymorphic transformation.

i.b. When the glass transition temperature (Tg) is lower than the Tmill, the amorphous phase has too short a lifetime to be detected during the polymorphic transformation.

ii. According to the literature, most polymorphic transformation induced by milling exhibits sigmoidal kinetics characterized by a striking incubation period during which the initial form is preserved, followed by a quite fast transformation stage leading to the final form [9,10,11,12,13,14]. The long metastability of the initial form during milling is puzzling, and it has been proposed that the transformation is triggered when a specific microstructure of the powder is reached [9,10,12,14].

iii. The duration of the transformation and, therefore, the shape of the kinetics can depend on the intensity of the milling carried out [15].

iv. This mechanism seems to be valid for both monotropic and enantiotropic crystalline forms.

To the best of our knowledge, this overall pattern of transformation is quite consistent with all of the polymorphic transformations induced by milling already described in the literature (see the corresponding review in this issue [9]). However, examples of such transformations remain scarce and their kinetics have been very little studied. Expanding the number of well-characterized cases would therefore be highly valuable to further elucidate the microscopic mechanisms governing these transformations.

In this work, we present some new experimental results concerning the polymorphic transformation upon milling of several compounds: mannitol, famotidine, sulfamerazine, and glycine. They are expected to provide crucial information to assess the validity of the mechanism proposed from the previous literature analysis and to propose a general mechanism applicable to polymorphic transformations of pharmaceuticals materials induced by milling.

To discuss point i.a, milling experiments were conducted on famotidine (Tg = 50 °C) in order to unambiguously show the existence of an amorphous form intermediate or coexisting with the final form after milling. For this compound, the work of Lin et al. [16] shows that there is a polymorphic transformation B→A upon milling, but no information on a potential transient amorphous phase was given, although Tg is higher than Tmill. This aspect is further investigated in the present study. To discuss point i.b, the milling of mannitol, a material with a Tg below room temperature (Tg = 13 °C), was performed. This material has polymorphic transformation kinetics under milling that unambiguously shows a sigmoidal shape, as clearly evidenced by the work of Martinetto et al. [12]. The experimental work carried out here aims to prove the existence of a transient amorphous fraction during polymorphic transformation, even though the lifetime of amorphous material is too short to be observed directly.

To discuss point ii, the work carried out here on sulfamerazine aims to confirm the kinetic profile of the milling-induced transformation of a material whose passage through an intermediate amorphous form is very clear in the literature [17], in agreement with its Tg of 62 °C.

To discuss point iii, milling experiments were conducted on glycine. It is clear for this compound that the duration of the transformation depends on the intensity of the milling carried out [15], and that the transformation kinetics are not well characterized.

The polymorphic forms involved in the milling-induced polymorphic transformations are characterized by an enantiotropic relationship in the case of glycine and sulfamerazine, and are characterized by a monotropic relationship in the case of famotidine and mannitol. These different relationships will also allow us to discuss point iv.

## 2. Materials and Methods

### 2.1. Milling

Two types of mills were used: a planetary mill and a vibrating mill.

The planetary mill used was the Pulverisette 7 marketed by Fritsch (Pittsboro, NC, USA). It consists of a disk on which two milling jars with a capacity of 45 mL are fixed. These jars are used with 7 milling balls of 15 mm diameter and 1 gram of samples, leading to a ball/sample mass ratio of 75/1. These parameters are those recommended by the manufacturer (Fritsch) for optimal milling. The jars and the balls are made of zirconium oxide, a material known for its resistance to shocks and high wear, thus allowing their use for a prolonged milling duration. The rotation speed of the disk and that of the jar was set to 400 rpm. The milling operations were carried out at room temperature (RT: approximately 20 °C). The milling periods (typically 20 min) were carried out alternately with pause periods (typically 10 min) in order to avoid heating the sample during long millings. Heat-sensitive stickers on the milling jar indicated that the temperature of the jar reached approximately 35 °C under the given conditions.

The vibrating mill used was the MM400 marketed by Retsch (Haan, Germany). It consists of two vibrating arms, each carrying a milling jar with of 10 mL made of zirconium oxide. One gram of sample was used for each milling experiment, with a 15 mm zirconium milling ball, and the milling frequency was set to 30 Hz. The milling periods (typically 10 min) were carried out alternately with pause periods (typically 5 min) in order to avoid heating the sample during long millings.

### 2.2. Laboratory X-Ray Diffraction

The diffractometer used was a Panalytical Xpert Pro (Malvern Panalytical Ltd., Malvern, UK) equipped with a copper anticathode tube that produces an X-ray beam with a wavelength of λ_1_ = 1.5405 Å, λ_2_ = 1.5440 Å, and an Xcelerator detector (Malvern Panalytical Ltd., Malvern, UK). The X-ray diffraction (XRD) experiments were carried out on a wafer system, i.e., a metal sample holder hollowed out in its center. A sufficient quantity of material to fill the cavity and obtain a surface of material without relief is placed there. The wafer is rotated to limit the effects of preferential orientations and to maximize the number of grains exposed to the beam. For each series of analyses, the diffractometer is calibrated with a sample of NAC (Na_2_CaAl_2_F_14_), which is characterized by very thin and isolated Bragg lines on the angular range 12–102° in the scale of 2θ. This calibration is mainly used for microstructural analysis, with the Bragg lines of the NAC being considered to reflect instrumental resolution on a very wide angular range. The acquisition is conducted on an angular range from 5 to 60° in 2θ at a rate of 50 s per point with a step of 0.0167°.

The Rietveld method was used to perform the structural analysis for each diffraction diagram. The analyses were carried out with MAUD (Version 2.992) [18] software.

The quantification of the crystallite and amorphous phases in a crystallite mixture was also achieved using the Rietveld method. The fraction of crystalline phase in the mixture affects the intensity of the diffraction peaks. An external standard with known proportions mixed with the sample of the milled material can be used to calculate the percentage of amorphous and crystalline phases (external standard method, Equation (1)) [19]:(1)$$W_{a}=1-\frac{W_{s}}{W_{s,calc.}}$$

$W_{a}$ is the percentage of amorphous phase, $W_{s}$ is the percentage of the standard, and $W_{s,calc.}$ is the percentage of standard calculated by the Rietveld method. Here, 10 mass % of silicon was used as a standard added to the milled compound.

### 2.3. Synchrotron Experiments

Figure 1 shows the setup that was used on the high-resolution ID22 [20] beamline of the European Synchrotron Radiation Facility (ESRF, Grenoble, France) to perform the in situ milling experiments (experiment number HC-4992). This setup was inspired by what was in place on ID15B for the mannitol [12] experiments. An MM400 mill, the equivalent of a setup used in the laboratory, was used. The MM400 mill used on ID22 was drilled so that the X-ray beam entered from the back of the mill (see Figure 1 left) and passed through it to reach the plexiglass jar in which the sample was placed in powder form. This type of transparent jar allows the beam to pass through and reach the sample. The X-rays were then diffracted by the sample and collected by the detector placed in front of the mill (see Figure 1 right). The recording of the diffraction patterns could therefore take place during the milling phase itself.

Moreover, ID22 is equipped with a multi-analyzer stage (thirteen Si analyzer crystals positioned every 2° in 2θ) mounted on a 2θ arm. The 2θ arm is scanned at a particular speed and thirteen patterns are collected simultaneously, before averaging. A silicon sample (Si NIST 640c) with a unit cell parameter equal to 5.43119 Å was used for calibration. The selected energy was 35 keV and corresponds to a calibrated wavelength of λ= 0.354330 Å. Data were collected at room temperature in the 0.5–27° 2θ range at a speed of 5°/min.

### 2.4. DSC Experiments

In this work, a DSC Q200 from TA instruments (New Castle, DE, USA) was used. During all measurements, the calorimeter head was flushed with highly pure nitrogen gas. Temperature and enthalpy readings were calibrated using pure indium at the same scan rates as those used during the measurements of the samples. The sample mass used was a few milligrams (less than 6 mg). Samples that were not subject to sublimation were placed in standard open crucibles (aluminum capsules without lids). Those that sublime at high temperature were, on the other hand, placed in hermetic aluminum crucibles.

### 2.5. Materials

The glycine (C_2_H_5_NO_2_) used for the study was supplied by SIGMA (St. Louis, MO, USA). The product was more than 99% pure and was in the γ form, according to the supplier.

Sulfamerazine (4-Amino-N-(4-methyl-2-pyrimidinyl)benzenesulfonamide: C_11_H_12_N_4_O_2_S) was purchased from SIGMA Aldrich (purity higher than 99%) and used without further purification. The commercial compound is the form I.

Mannitol (C_6_H_14_O_6_) was purchased from Fluka (Geel, Belgium) (purity higher than 99%) and was used without any further purification. The commercial compound is the stable polymorph β.

Famotidine (C_8_H_15_N_7_O_2_S_3_) was purchased from TCI (purity higher than 98%) and was used without any further purification. The commercial compound is the form B.

## 3. Results

### 3.1. Mannitol

Figure 2 shows the X-ray diffraction patterns of crystalline mannitol recorded before milling (in black) and after a 3 h milling process (in blue). The two diffractograms are clearly different, indicating that milling has induced polymorphic transformation from form β to form α, as already reported in reference [12]. So far, no sign of transient amorphization could be detected during the polymorphic transformation of mannitol, even through in situ power X-ray diffraction allowing the real-time monitoring of the structural state. This is likely due to the recrystallization of amorphous mannitol, which is known to be very fast [21]. In order to highlight a potential transient amorphization step during the β→α polymorphic transformation of mannitol, 65% of the latter was mixed with 35% of lactose previously amorphized by milling [22]. Figure 2 also shows the X-ray diffraction pattern of this physical mixture recorded after a 12 h milling process. We can observe small Bragg peaks characteristic of the form α of mannitol superimposed on a broad diffusion bump, revealing the presence of an amorphous fraction. This indicates that the 12 h milling process has produced a heterogeneous sample made of amorphous mannitol–lactose alloy and crystalline mannitol α. It must be noted that no further changes in the X-ray diffraction pattern could be detected for longer milling times, indicating that a stationary structural state upon milling has been reached. The Cp jumps of the physical mixture recorded by DSC before and after the co-milling are shown in the inset of Figure 2. They show a strong shift of Tg toward the low temperature as well as a noticeable increase in the Cp jump amplitude at Tg. Both features signal the molecular dispersion of mannitol molecules into the initial amorphous lactose. This increases the global amorphous fraction in the sample and induces the plasticization of amorphous lactose by mannitol.

The transformation kinetics of the [65:35]*w*/*w* mixture was monitored by sampling the material after different milling times and by analyzing it by X-ray diffraction and DSC. Figure 3 shows the evolution of the structural composition of mannitol in this mixture during milling. For each milling time, the amorphous fraction (Xam) was quantified by comparing the amplitude of the specific heat (Cp) jump at Tg in the DSC scan to that of the fully co-amorphized material, while the crystalline α and β fractions (respectively Xα and Xβ) were quantified by X-ray diffraction. These two quantification methods are completely independent and lead to different uncertainties (which can be quite high in the case of measuring the jump of Cp at Tg). That is why the sum of the fractions may slightly differ from 1. Nevertheless, as shown in Figure 3, the structural evolution can be clearly divided in two stages. In the first stage, corresponding to the first 200 min of milling, 47% of the initial mannitol is co-amorphized with lactose, while no trace of mannitol α could be detected. In the second stage (after 200 min of milling), the remaining 53% of mannitol β is transformed into mannitol α while the co-amorphized fraction of mannitol remains constant.

These structural evolutions indicate that mannitol first undergoes amorphization upon milling, forming an amorphous solid dispersion with lactose that appears to be resistant to recrystallization due to the high glass transition temperature of lactose (Tg ≈ 110 °C) [22]. However, due to the plasticization effect of mannitol, the physical stability of the dispersion decreases as its mannitol concentration increases [23]. As a result, during the second stage of the evolution, lactose can no longer stabilize the newly amorphized mannitol, which then recrystallizes toward the form α. It thus appears that the polymorphic transformation β → α of mannitol upon milling is not direct and involves a transient amorphization stage leading to a transformation mechanism consistent with the transformation mechanism previously identified in sorbitol [10,14]. Such a transient amorphous state could be identified here thanks to its stabilization induced by the formation of a high Tg molecular alloy with lactose during the early stage of milling. The mechanism of the polymorphic transformation of mannitol upon milling can thus be rationalized using the microscopic model used to describe the transformation of sorbitol by Dupont et al. [10,14]. In this model, the polymorphic transformation under milling from the initial form to the final form of a compound is modeled by two successive structural changes induced by each mechanical impact: an amorphization of the impacted crystallites immediately followed by recrystallization of the corresponding amorphous grains. Moreover, the nature of this recrystallization (toward the initial form or toward the final form) depends on the structural state of the adjacent crystallites. In particular, it is strongly directed toward the predominant crystalline form surrounding the amorphous grain. This conditional recrystallization leads, in the early stages of milling, to a stationary state consisting of a small fraction of the crystallites of the final polymorphic form dispersed among a multitude of crystallites of the initial form. This situation persists until a cluster of crystallites of the final form is incidentally obtained. The crystallites of the final form involved in this cluster are much more stable than the isolated crystallites because each of them is in contact with several crystallites of the final form. Once such a stable cluster is established, it promotes the recrystallization of adjacent amorphous regions into the final form, leading to a rapid overall polymorphic transformation.

In the case of mixture (lactose/mannitol studied here, or HCT/sorbitol [14]) the situation is slightly different from that of pure compounds [10,24]. This difference appears through a much slower kinetics of transformation in the mixture compared to pure compounds. It is characterized by a longer incubation time, for example, 200 min in the mixture of lactose/mannitol (see Figure 3) instead of 30 min in the pure compound (see reference [24]) and 8 h in HCT/sorbitol (see reference [14]) instead of 2 h in the pure compound (see reference [10]). Moreover, a longer transformation stage is also observed. It is, for example, 400 min in the lactose/mannitol mixture (see Figure 3) instead of 100 min in the pure compound (see reference [24]) for 50% of transformation. The same results are obtained in the case of sorbitol (10 h in the HCT/sorbitol mixture [14] instead of less than 1 h in pure compounds [10]). Both slowing effects can be inferred from the mechanism of transformation proposed above. Due to the partial formation of an amorphous lactose/mannitol alloy in the very early stages of co-milling, the mannitol α-crystallites are dispersed among amorphous alloy grains and a few β-crystallites. This greater dispersion significantly reduces the probability of forming a stable cluster of α-crystallites, which inevitably results in an increase in the induction time required for its formation. Moreover, the presence of stable amorphous grains at the periphery of the stable cluster then decreases its growth rate. The presence of a permanent fraction of amorphous grains in the system is therefore at the origin of a greater isolation of the α crystallites, which delays the formation of a stable cluster of α crystallites and slows down the subsequent β→α transformation. This globally slower kinetics of transformation of the mixture compared to that of pure mannitol thus further proves the reality of the transformation mechanism we propose.

### 3.2. Famotidine

To observe the behavior of famotidine under milling, a sample of commercial famotidine was milled for 11 h at room temperature in a planetary mill. This sample was analyzed before milling and after 2 and 11 h of milling using XRD. The corresponding diffraction patterns are shown in Figure 4. The diffraction pattern recorded before milling shows the Bragg peak corresponding to the commercial B form of famotidine [16]. No diffusion halo characteristic of an amorphous fraction is observed. The diffraction pattern of 2 h milled famotidine still shows the Bragg peaks of form B; however, these are broader and much less intense than before milling. Moreover, the presence of an underlying halo suggests the formation of a noticeable amorphous fraction. A few peaks of the form A of very low intensity can also be detected on the diffractogram, which show that the polymorphic transformation is just beginning. The diffraction pattern of the 11 h milled famotidine is characterized by the Bragg lines of the form A and the total disappearance of the diffusion halo and the Bragg peaks characteristic of form A. This indicates that a total polymorphic transformation B → A has occurred during the milling process. No further structural changes could be observed for longer milling. The previous evolution clearly indicates the amorphization of a part of famotidine on the one hand and the existence of an induction time prior to transformation for this material on the other.

These observations on famotidine confirm the transformation mechanisms mentioned in the literature review [9] and recalled in the introduction for a monotropic relationship between two polymorphic forms. For a material with a Tg slightly higher than room temperature, we directly observe both the partial amorphization of the compound and an induction time in the polymorphic transformation due to the amorphization/recrystallization mechanism.

### 3.3. Sulfamerazine

An investigation by Macfhionnghaile et al. [17] of sulfamerazine shows that after 60 min of room temperature vibrational milling (at 25 Hz), sulfamerazine form I disappears completely and results in a mixture of 78% of form II and 22% of amorphous phase. Additionally, further milling does not induce any changes. The existence of an amorphization stage in the mechanism of polymorphic transformation induced by milling is thus obvious in this case. We can thus wonder whether this amorphization is linked to the sigmoidal kinetics of the polymorphic transformation showing an induction time.

The case of sulfamerazine is of particular interest because milling-induced polymorphic transformations between enantiotropically related forms are documented very little in the literature [9]. We have thus reinvestigated the effect of vibrational milling (30 Hz) on sulfamerazine to determine the kinetics of its polymorphic transformation. Different samples of sulfamerazine were milled for time periods varying from 0 to 60 min. At the end of each milling experiment, the structural state of the sample was analyzed by powder X-ray diffraction using an aluminum plate sample holder and by DSC (5 °C/min) using a hermetic aluminum pan. The results of the analysis are shown in Figure 5, where the dashed lines mark the peaks corresponding to form I (black lines at, e.g., 20.14°, 22.32°) and form II (red lines at, e.g., 15.44°, 21.14°, 24.62°) of sulfamerazine. It is clear that the X-ray diffraction pattern recorded before milling (0 min) only shows the Bragg peaks characteristic of form I, while that recorded after 60 min of milling only shows the Bragg peaks characteristic of form II. This clearly indicates that a complete phase transformation I → II occurred in the course of this 60 min milling process.

The evolution of the X-ray diffraction pattern and thermogram reported in Figure 5 also shows that the evolution can be divided in two stages.

From 0 to 30 min of milling (stage 1), the Bragg peaks characteristic of form I strongly decrease, while those characteristics of form II can hardly be detected. This evolution suggests a partial amorphization of form I. We also note a noticeable broadening of Bragg peaks of form I at the very beginning of the milling signaling and a strong reduction in the size of the crystallites induced by the mechanical shocks. Moreover, the quantification from the X-ray diffraction diagrams indicates that form I content falls from 100 to 80%, being replaced by amorphous content. The thermogram recorded after 25 min of milling shows an exotherm ranging from 40 to 70 °C, which confirms this partial amorphization of the material. It must be noted that the glass transition of the amorphous fraction is not observed and that its recrystallization starts below the expected Tg (62 °C) of sulfamerazine [17]. Such behavior is not unusual and was already observed in several materials, with one of the most notable cases being griseofulvin [25]. In that case, it has been demonstrated that the unusually high molecular mobility of the milling-induced amorphous material allows the triggering of the recrystallization a few degrees below Tg. Such depressed recrystallization can totally mask the glass transition signature.

From 30 to 40 min of milling (stage 2), The Bragg peaks of form I further decrease and vanish while new Bragg peaks characteristic of form II suddenly develop. They are likely to come from the recrystallization of the amorphized fraction. The quantification indicates that form I content dramatically decreases until 0%, while form II content increases from 0 to 100%, replacing form I and the amorphous form, which decrease from 20 to 0%. This total transformation toward form II is confirmed by the heating (5 °C/min) DSC scan of sulfamerazine milled for 60 min, which clearly shows the endotherm characteristics of the polymorphic transition II → I at 135 °C. No further evolution of either X-ray diffractograms or thermograms could be observed for longer milling times, indicating the completion of the transformation.

The time evolution of the structural components’ fractions during milling were determined via PXRD analysis and are reported in Figure 6. There is a clear induction time before a fast transformation I → II giving rise to sigmoidal kinetics. The figure shows that the transformation is mediated by a large fraction (up to 20%) of amorphous sulfamerazine developing transiently during the incubation time. This stationary amorphous fraction during the incubation time reveals competition between the amorphization induced by the mechanical shocks and a recrystallization of the amorphized fractions between two shocks. This dynamic equilibrium breaks when the faster recrystallization toward form II starts, making the transient amorphous fraction hardly detectable. These observations on sulfamerazine also confirm the transformation mechanisms emerging from the literature review [9] for a enantiotropic relationship between two polymorphic forms this time. Moreover, these results are clearly in agreement with those of Macfhionnghaile et al. [17] where form I of sulfamerazine is transformed into a mixture of amorphous form (20%) and form II (80%) after 60 min of milling.

### 3.4. Glycine

Glycine is a compound that has three known polymorphic forms—α, β, and γ [26]—and shows a γ → α polymorphic transformation upon milling [15]. The polymorphic transformation kinetics of glycine was studied here by powder X-ray diffraction both ex situ in the laboratory and in situ on a synchrotron. With the exception of the kinetics of the polymorphic transformation of mannitol, the experiments described in the literature (see reference [9]) were all carried out ex situ and thus present some limitations. Indeed, this type of experiment requires interrupting the milling, taking a portion of the powder, and placing it in a sample holder (wafer or Lindemann capillary) suitable for XRD. As a result, the sample is then analyzed at least a dozen minutes after the milling has stopped. In addition, the X-ray diffraction analysis itself also lasts a few dozen minutes. These two points are problematic, as structural evolution of the material could occur during both the preparation and measuring time of the sample to be analyzed. In particular, this is crucial during the induction time usually preceding the beginning of the polymorphic transformation where the reversion tendency toward the initial crystalline form is high. In synchrotron in situ analysis (i.e., during the milling itself), it is thus highly desirable to shorten the time required for the analysis as much as possible in order to overcome the tendency for the material to return to its initial state in the absence of milling. In addition, the different phases are sometimes present in very small quantities so that their detection and their precise evolution over time from the first moments require highly sensitive instruments and an in situ analysis (i.e., during the milling itself). This is why, for a better understanding of the transformations upon milling, we compared the kinetics of the polymorphic transformation of glycine determined by powder X-ray diffraction during ex situ and in situ milling experiments. For the latter, we used synchrotron X-ray diffraction (ESRF ID.22) to follow the structural evolution of the material during milling in real time as this approach has high sensitivity to detect small quantities of material.

#### 3.4.1. Ex Situ Experiments

For ex situ experiments, a sample of commercial glycine (1.1 g) was milled at room temperature for 20 h in a planetary mill. The milling was regularly stopped to analyze the structural composition of the milled powder by XRD. The XRD analyses were carried out in wafers. Then, 500 mg of material was taken from the jar, analyzed by XRD, and then replaced in the jar to continue milling. This protocol makes it possible to maintain a constant quantity of glycine in the jar during milling. The evolution of the diffractograms for milling times ranging from 0 to 1200 min is shown in Figure 7.

Unmilled glycine only exhibits the Bragg peaks of the γ form of glycine [27]. As milling progresses, a significant change in the diffraction pattern is observed. In particular, we note the progressive appearance of the Bragg peaks characteristic of the α form (e.g., at 2θ = 19° and 24°) in parallel with a disappearance of the peaks of the initial γ form (e.g., at 2θ = 22 and 25°). We note that the peaks characteristic of the form α increase in intensity as the milling time increases, while peaks of the γ form decrease. The peaks of the γ form have completely disappeared after 1200 min and no trace of amorphous material can be detected, indicating total polymorphic transformation. The diffractograms recorded during milling were analyzed via the Rietveld method in order to quantify the fraction of each polymorphic form in the sample. The temporal evolution of the α-form fraction during milling is reported in Figure 8.

The transformation kinetics appear to be fast, with 90% of the initial γ form converted into the β form after only 80 min of milling. Moreover, the kinetics show an overall slightly sigmoidal shape with slight acceleration during the first 10 min of milling. No notable induction period is observed, which constitutes an essential difference with the transformation kinetics of sorbitol [10], mannitol [12], sulfamerazine, and almost all polymorphic transformations of pharmaceutical compounds available in the literature [9].

#### 3.4.2. In Situ Experiments

The in situ experiments were performed at ESRF on the beam line ID22, which made it possible to follow the transformation accurately and almost in real-time conditions. The adaptation of a setup already developed for the in situ analysis of milled materials was necessary (see Experimental Section 2.3 for details). In previous experiments, a plexiglass milling jar had been used on the ID15B line to study the kinetics of mannitol transformation [12]. These experiments showed that the sensitivity of the measurement was sufficient to detect small quantities of material present from the start of milling, as well as the possibility of obtaining relevant information on short times in terms of both structural composition and microstructure (crystallite size, amorphous phase). In practice, a diffraction pattern was recorded every minute, allowing the evolution of the kinetics to be accurately monitored. However, due to an insufficient signal-to-noise ratio, seven diffraction patterns were averaged (~7 min of milling time) to obtain a well-defined pattern, allowing for the quantification of the different polymorphic phases γ and α thanks to Rietveld analysis (see Figure 9).

From the X-ray diffraction patterns, the kinetics of glycine transformation during the in situ milling was obtained and is reported in Figure 10. The figure shows striking differences with that recorded during the ex situ milling experiment (Section 3.4.1). The kinetics show, in particular, a sigmoidal shape with a long incubation time of about 200 min. We also note that the timescale of the transformation is not the same as in the ex situ experiment (Figure 8) since the transformation of more than 90% of glycine requires 800 min of milling compared to 80 min in the laboratory (see Figure 8 and Figure 10). Such differences in duration can be explained by several parameters (different number of balls, jar made of zirconia in the laboratory and plexiglass on synchrotron, different milling technology, synchrotron radiation intensity), which makes the intensity of in situ milling weaker than that used for ex situ milling. Moreover, this is coherent with the results of Matsuoka et al. [15] indicating that the transformation time of glycine is longer when the milling intensity is lower.

These observations reflect sigmoidal kinetics with a fairly long induction time compared to the transformation time, a relatively rapid transformation, and a non-negligible seeding effect of the γ form. These observations are all consistent with the proposed model, even if it was not possible to observe the formation of the amorphous phase in situ. This behavior is most likely attributable to the explosive crystallization tendency of glycine, which renders the lifetime of its amorphous state extremely short. To date, and to the best of our knowledge, no amorphization technique has succeeded in producing amorphous glycine. Furthermore, our attempts to co-amorphize glycine with high-Tg amorphous excipients by co-milling were unsuccessful, probably because amorphous glycine recrystallizes before it can be dispersed within the excipient matrix.

## 4. Discussion

The results presented here enable the completion of the summary table of polymorphic transformations of pharmaceutical materials induced by milling previously reported in a recent review [9]. The updated version of this table is provided as Appendix A. These new findings further support the mechanism of milling-induced polymorphic transformations proposed in the review, which was initially inferred from a quite limited set of examples.

The results reported here on mannitol, famotidine, sulfamerazine, and glycine are in line with the recent bibliographic analysis carried out for some components, especially bezafibrate, chloramphenicol palmitate, and indomethacin, and contribute to a better understanding of the mechanisms that govern the polymorphic transformations of molecular materials in the solid state under mechanical milling. They confirm, in particular, that the polymorphic transformations under milling have the following characteristics:-The existence of an induction time followed by a rapid transformation from the initial form to final form.-The transformation of part of the material (permanently or temporarily) into an amorphous material (except for glycine, for which any amorphization could be detected). It should be noted that the observation of transient amorphization could sometimes only be carried out indirectly.

These observations are linked to the hypothesis that transformations under milling are likely to result from competition between an amorphization process due to mechanical shocks and thermally activated recrystallization [9]. To go further, it is shown here that the recrystallization rate depends on the relative position of the Tmill and Tg for a given pharmaceutical compound. The most extreme case, where the Tg is much higher than the milling temperature, is well known; this is where the material is amorphized without ever recrystallizing [6,28,29]. Then, when the gap between the Tg and Tmill decreases, the molecular mobility of the amorphized material allows for the partial recrystallization of this amorphous material. It is clearly shown that when the Tg is several tens of degrees higher than the milling temperature, a transformation of the initial form into a mixture of amorphous form and final form is observed. This is clearly observed in the context of this work on sulfamerazine (Tg = 62 °C, see Figure 6) and famotidine (Tg = 50 °C, see Figure 4) but also in previous work on sulfathiazole [30] (Tg = 67 °C), indomethacin [31] (Tg = 47 °C), and cimetidine [32] (Tg = 43 °C). When the Tg approaches the milling temperature, a larger fraction of the amorphous material recrystallizes significantly faster compared to the previous case, so that the presence of the amorphous material serving as an intermediate for the transformation is barely detectable. This phenomenon is observed on bezafibrate [11] (Tg = 40 °C), ranitidine hydrochloride [33] (13 < Tg < 30 °C), rivastigmine hydrogen [34] tartrate (Tg = 38.2 °C), and fananserine [8] (Tg = 19 °C). For these materials, close monitoring (experiments carried out over several milling times and several transformation rates from the initial form to the final form) allows the presence of the intermediate amorphous form to be observed. However, at the end of milling, this amorphous fraction is no longer detectable because it has completely recrystallized. Finally, when the milling temperature is higher than the Tg, the direct observation of the intermediate amorphous material is quite impossible, which is confirmed here thanks to the work on mannitol (Tg = 13 °C; see Figure 2 and Figure 3) and that is correlated with the observations made on sorbitol [10,14] (Tg = −3 °C). In this case, the transient amorphous state needs to be stabilized to be observable. This was achieved here by mixing mannitol with a high-Tg amorphous excipient.

The competition between amorphization and recrystallization, combined with preferential recrystallization toward the predominant form, can potentially lead to two dynamic microstructural equilibrium states. The first one corresponds to a few crystallites of the final form dispersed within a myriad of crystallites of the initial form, while the second one represents the opposite configuration. Dupont et al. [10,14] demonstrated that the transition between these two states arises from the accidental formation of a cluster of crystallites of the final form (see Section 3.1 for details). This mechanism results in sigmoidal transformation kinetics, with an induction period corresponding to the time required for the cluster formation. In our results, these sigmoidal kinetics are effectively observed in the case of mannitol (reference [12] and Figure 3), sulfamerazine (see Figure 6), and glycine (see Figure 10). This is also observed on other compounds in the literature [9] such as sorbitol [10,14], bezafibrate [11], ranitidine hydrochloride [33], rivastigmine hydrogen tartrate [34], indomethacin [31], fananserine [8], and chloramphenicol palmitate [13] (see Figure 11 for some examples).

Figure 11 compiles most of the polymorphic transformation kinetics reported in this study as well as in the literature. For the sake of comparison, each kinetic value has been rescaled with respect to the half-transformation time. Interestingly, all kinetics, except those obtained through in situ milling experiments, overlap reasonably well given the very different milling conditions and analysis techniques. This scaling behavior thus strongly suggests that milling-induced polymorphic transformations are governed by a universal mechanism. For the kinetics obtained by in situ milling using a synchrotron, it is possible that the perturbation induced by the very intense synchrotron radiation [35] combines with that due to milling to provide a more complex mechanism of transformation.

Moreover, it is noteworthy that the polymorphic forms involved in the transformation of sorbitol, mannitol, ranitidine hydrochloride, rivastigmine hydrogen tartrate, famotidine, indomethacin, fananserine, and cimetidine have a monotropic relationship, while those involved in the transformation of glycine, bezafibrate, and sulfamerazine have an enantiotropic relationship. In the case of monotropically related forms, the laws of thermodynamics do not predict any phase transition between the two forms upon heating or cooling. Only an irreversible transformation from the less stable form to the more stable form can possibly be observed. It is therefore difficult to envisage that polymorphic transformation induced by milling could occur directly between two monotropic forms without the occurrence of a transient amorphization step. Conversely, for an enantiotropic relationship between a pair of polymorphs, there is a thermodynamic transition point between the two polymorphs below the lowest melting temperature so that direct transition could be expected upon cooling or heating. It is therefore possible that milling-induced polymorphic transformations could also be direct. Interestingly, the results presented in this paper indicate that this is not the case and that the nature of the relationship between two forms does not appear to fundamentally affect the mechanism of polymorphic transformations induced by milling. In particular, a transient amorphization stage is clearly detected in both cases.

## 5. Conclusions

In this work, we investigated the kinetics of polymorphic transformations of mannitol, famotidine, sulfamerazine, and glycine under milling, with the aim of complementing a recent bibliographic analysis dedicated to such solid-state transformations [9]. Since these kinetics directly reflect the underlying microscopic mechanisms, they are expected to provide insights into these mechanisms that remain poorly understood. In particular, our results confirm three key features: the systematic sigmoidal shape of milling-induced polymorphic transformation kinetics, the presence of a significant induction period preceding the effective transformation, and the occurrence of an intermediate amorphous phase between the initial and final crystalline forms. All of the results agree with the work of Dupont et al. [10,14], which suggests that the polymorphic transformations under milling are triggered by the accidental formation of a cluster of crystallites of the final form. Moreover, a scaling analysis of kinetics present in this study and in the literature provides strong evidence that milling-induced polymorphic transformations are governed by a universal mechanism independent of both milling conditions and relationships (monotropism or enantiotropism) between initial and final forms.

## Figures and Tables

**Figure 1 pharmaceutics-17-01404-f001:**
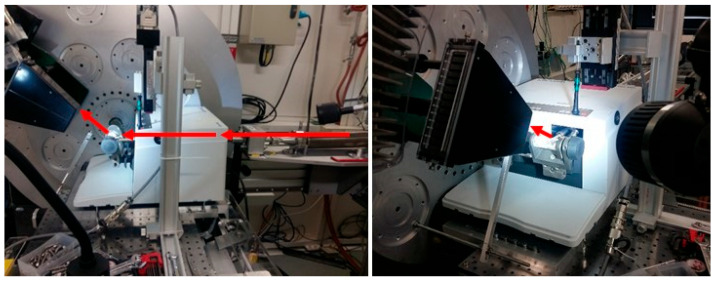
High-resolution diffraction setup for in situ milling (ESRF, ID22). (**Left**): Side view of the MM400 mill and detector. (**Right**): Three-quarter face view of the MM400 mill. The plexiglass milling jar is filled with glycine and a zirconia ball. The arrows represent the path taken by the X-rays.

**Figure 2 pharmaceutics-17-01404-f002:**
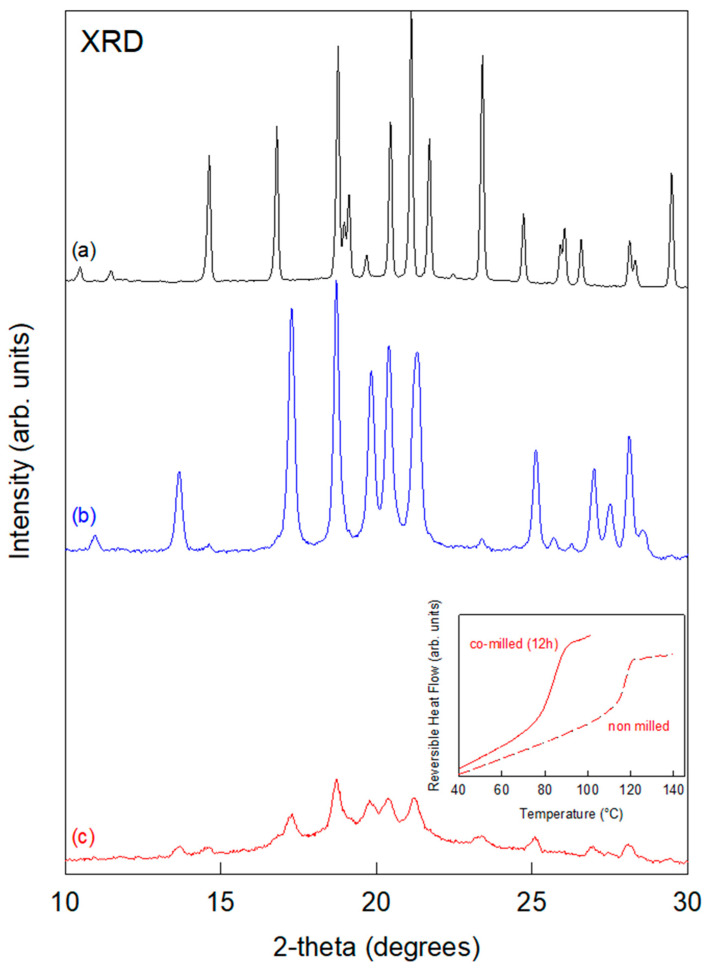
X-ray diffraction patterns recorded at RT: (**a**) crystalline mannitol before milling, (**b**) crystalline mannitol after 3 h of milling, and (**c**) physical mixture crystalline mannitol/amorphous lactose [65:35]*w*/*w* after 12 h of co-milling. The insert shows the reversible heat flow scans of the physical mixture (crystalline mannitol/amorphous lactose [65:35]*w*/*w*) recorded before and after a 12 h milling process.

**Figure 3 pharmaceutics-17-01404-f003:**
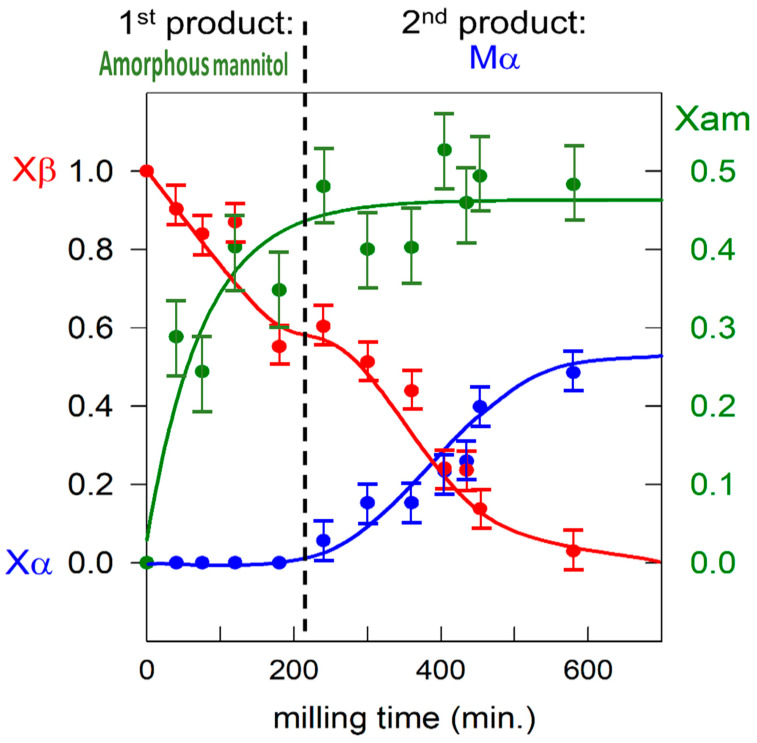
Time evolution of the fractions of each form of mannitol (α, β, and amorphous) during the co-milling of a mixture of 65% mannitol and 35% amorphous lactose.

**Figure 4 pharmaceutics-17-01404-f004:**
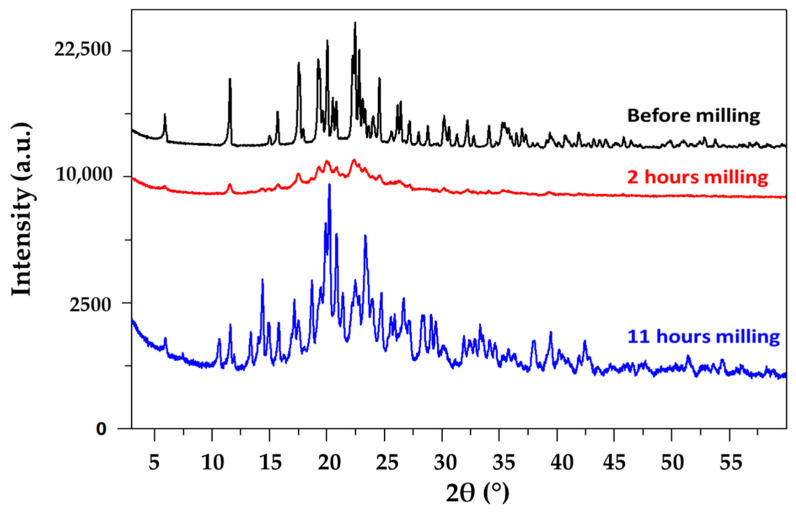
X-ray diagram of famotidine form B recorded before milling (black, top), after 2 h of milling (red, middle), and after 11 h of milling (blue, bottom). In this latter case, the X-ray diffraction pattern is characteristic of form A.

**Figure 5 pharmaceutics-17-01404-f005:**
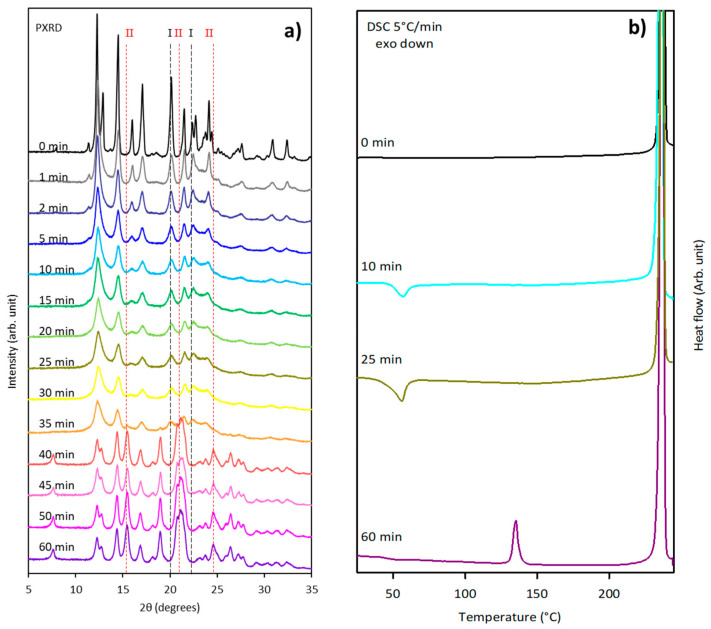
(**a**) X-ray powder diffraction patterns of sulfamerazine form I recorded after different milling times varying from 0 to 60 min. Milling times are reported on the left-hand side of the figure: black and red vertical lines mark the positions of some typical Bragg pics of forms I and II, respectively. (**b**) DSC curves of sulfamerazine recorded upon heating (5 °C/min) after different milling times ranging from 0 min to 60 min. The milling times are reported on the left-hand side of the figure.

**Figure 6 pharmaceutics-17-01404-f006:**
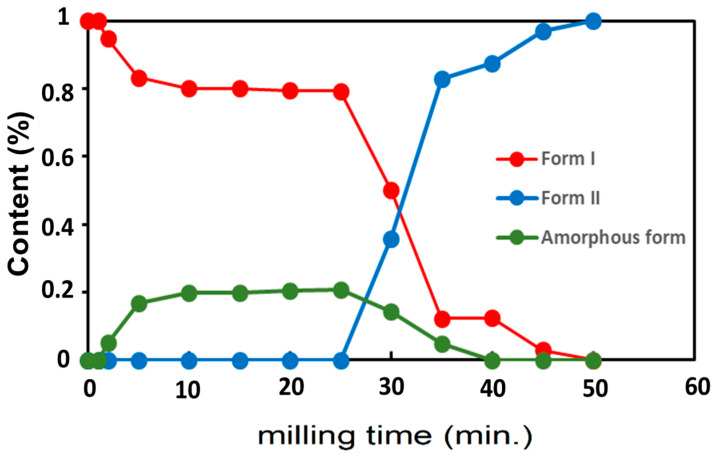
Time evolution of the structural composition of sulfamerazine during milling.

**Figure 7 pharmaceutics-17-01404-f007:**
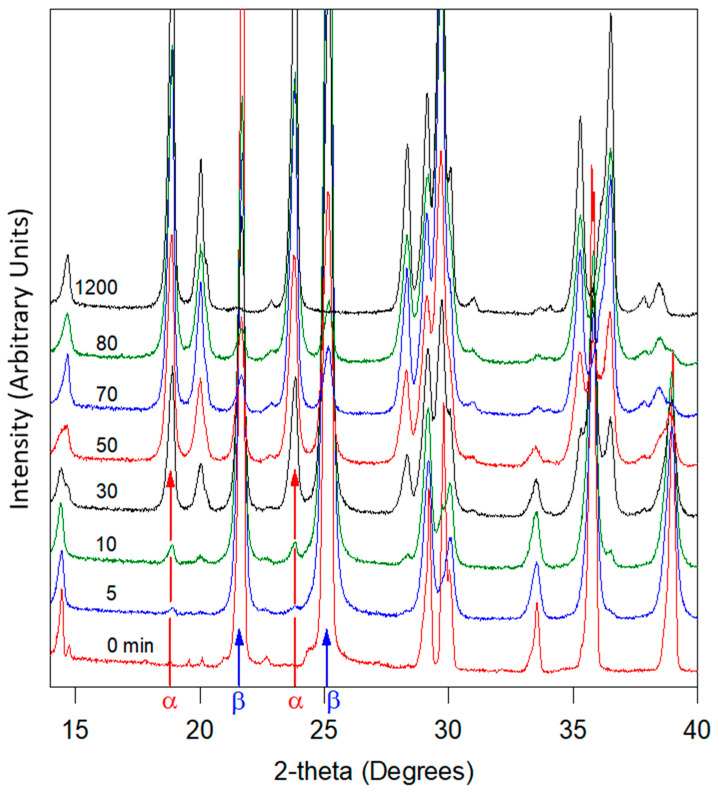
Diffractograms of glycine recorded at room temperature after different milling times. Milling times are indicated on the left-hand side of each diffractogram. The colors are used to better separate the diffractograms in case of overlapping. Arrows mark the main peaks characteristic of the α form and the γ form.

**Figure 8 pharmaceutics-17-01404-f008:**
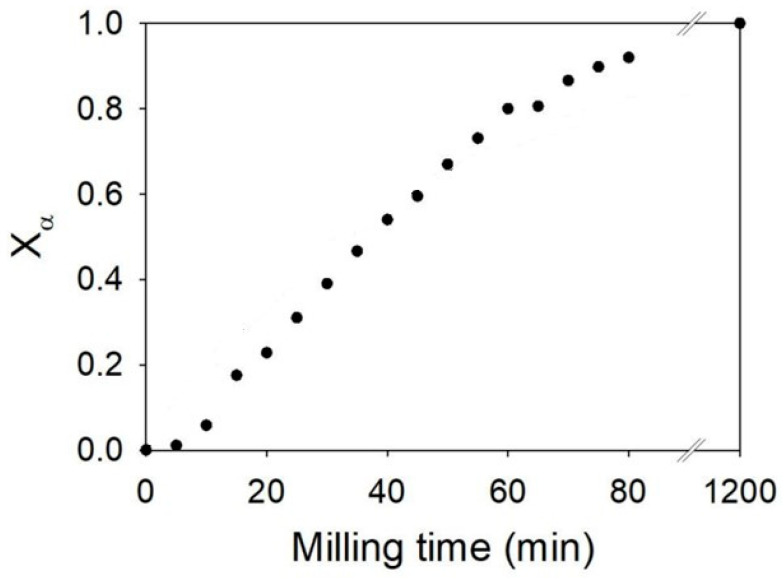
Kinetics of transformation γ → α of glycine obtained during ex situ milling.

**Figure 9 pharmaceutics-17-01404-f009:**
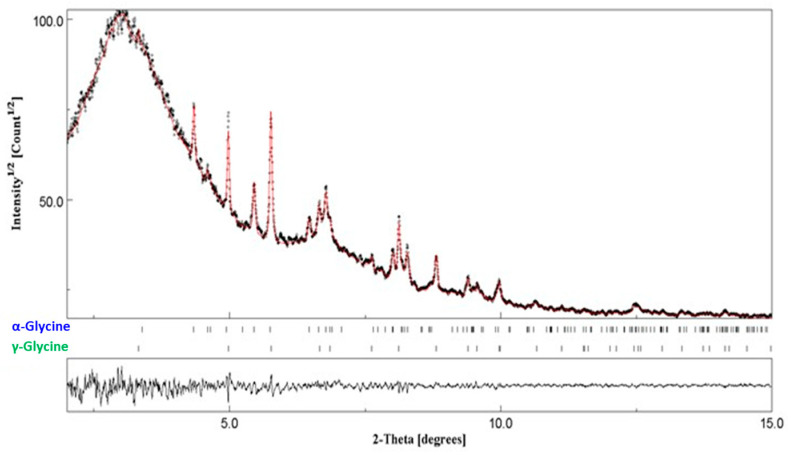
Rietveld refinement against the diffraction pattern of glycine obtained after 7 h of in situ milling (black, data points; red, calculated pattern). The high background noise at the beginning of the pattern comes from the plexiglass of the milling jar.

**Figure 10 pharmaceutics-17-01404-f010:**
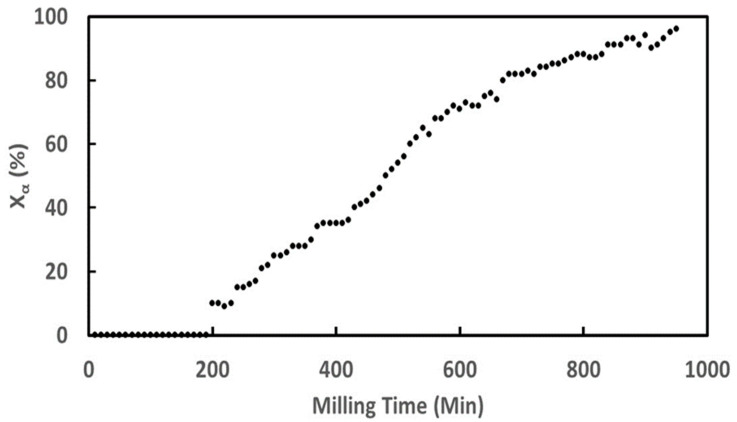
Kinetics of transformation of the γ phase to the α phase of glycine under milling carried out in situ.

**Figure 11 pharmaceutics-17-01404-f011:**
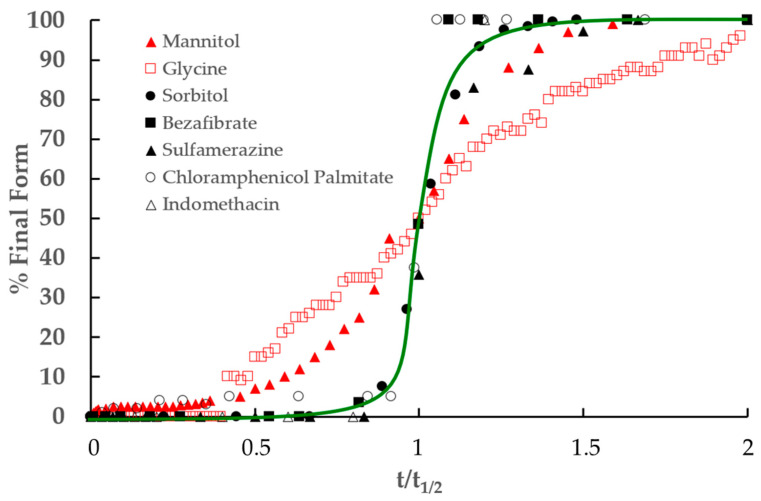
Kinetics of polymorphic transformations induced by milling for sorbitol, bezafibrate, sulfamerazine, mannitol, chloramphenicol palmitate, glycine, and indomethacin. These kinetics have been rescaled by their half completion time.

## Data Availability

Data are available on request.

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
