# Peer review of "New Kinetic Investigations to Better Understand the Mechanism of Polymorphic Transformations of Pharmaceutical Materials Induced by Milling"

_pharmaceutics, 2025, doi:10.3390/pharmaceutics17111404_

Round 1
Reviewer 1 Report
Comments and Suggestions for Authors
The manuscript investigates the kinetics and mechanisms of polymorphic transformations in several pharmaceutical materials (mannitol, famotidine, sulfamerazine, glycine) induced by milling. Using a combination of ex-situ and in-situ X-ray diffraction and DSC analyses, the authors show that these transformations generally proceed via a transient amorphous phase, with dynamics often following sigmoidal kinetics and including an induction period. The work supports a universal mechanism of milling-induced polymorphic transformations and provides comparative data across multiple compounds and experimental conditions.
The manuscript is overall scientifically sound and presents a clear experimental approach. The writing is generally understandable and manages to convey complex concepts effectively, although the English grammar and style require careful revision to meet publication standards. While the work delivers some new experimental insights, one central conclusion—that polymorphic transformations during milling typically proceed via an amorphous intermediate—is already an accepted concept in the field and does not at all represent a novel paradigm. Additionally, the manuscript contains redundancy/repetitive explanations; condensing these sections would improve readability without any loss of scientific value.
Was the temperature of the samples during milling monitored in all or at least in selected milling experiments? Please provide a statement in the manuscript. Milling frequently leads to warming of the sample, if the system is not cooled. Not controlling or monitoring the temperature during milling would be a certain design flaw of the study.
The authors repeatedly use the expression "...the fact that..." throughout the manuscript. The wording could either be a bit more careful or there should be better evidence (e.g. by references or further explanations) that it is a fact.
Comments on specific sections of the manuscript:
Title: The title of the manuscript does not match its scope as mannitol is not a therapeutic material. Anyway the experession therapeutic material is rather uncommon. Please resolve.
Line 28: Active priciple ingredient: Is Active Pharmaceutical Ingredient meant?
Line 34-35: "...because its polymorphic form both influences its physical stability and bioavailability." Please be more careful and add a "can" so that it reads "can influence its..." as this is not in every case true.
Line 64-76: This paragraph describes the mechanism of the solid form transitions during milling without providing any context. No literature is cited. Please explain at the beginning of this paragraph whether this is speculation by the authors (though reasonably) or cite the literature, where this is described.
Line 90: "works" remove s
Line 102: replace "have" by "has"
Line 229: replace "compound" as this a binary mixture. e.g. "material" could be used.
Figure 3: Please provide an explanation in the manuscript, why the sum of mol fractions are substantially different from 1 in some areas.
Line 259-269: This paragraph describes again a mechanism of solid form transition during milling and is redudant with lines 64-76 in the introduction section. Please have this mechanism described only once in the manuscript.
Line 280: "1 hours" please remove the s
Line 296: "...two 2 and 11 hours.." one 2 too much
Line 317: "two polymorphic form." Please use plural
Line 336: 2 grammar errors: Once a plural is used where the singulag should be used and vice versa.
Lines 339, 341: "from" instead of form
Figure 6: Please align style between Fig. 6 and Fig. 3
Figure 7: Due to the strong overlap of most reclections, I perceive the representation as confusing. Is it possible to scale and vertically shift the patterns in way that it becomes more clear?
Line 471: "scales" pl remove s
Line 522: "recrystallization mechanism". My understanding is, that it is rather the rate of recrystallization depends on the relative position of Tmill and Tg than the mechanism. Please replace mechanism by rate or explain why you think "mechanism is more appropriate.
Line 523: "for a given compounds" pl remove s
Line 532: Please use 3rd person singular verb form
Line 550ff: A mechanism of solid form transformation is oulined here for the 3rd time in manuscript which appears to me as an unnecessary redundancy. It is sufficient to have this described once in the manuscript.
Line 585-591: The authors state that the type of relationship between a pair of polymorphs (enantiotropy/monotropy) has "interestingly no fundamental impact on the transformation mechanism". This statement makes little sense to me as I don't see why this should or could be different. I recommend to remove this paragraph and instead discuss which of the observed transformations proceed at Tmill from the less stable to the more stable polymorphic form and which the other way.
Line 608-617: The mechanism of solid form transformation during milling is described here in detail for the 4th time in the manuscript. I think it, would be fully sufficient to have it described once in a manuscript section of choice. In this section, more context should be provided (references as well as better outlining the underlying evidence for the statements made)
Comments on the Quality of English Language
See "comments and suggestions for authors".
Reviewer 2 Report
Comments and Suggestions for Authors
This article, through the study of the crystal state changes of drugs such as mannitol, famotidine, and sulfamethoxazole under grinding, explains the connection between amorphization and polymorphism in the kinetic changes of drug crystal states, and adds practical examples to the review literature on the factors influencing drug crystal state changes. This article provides a detailed analysis of the kinetics of drug crystal forms and has certain research value. After minor revisions, it can be considered for publication. The following issues can be optimized and modified:
- How is Xam quantified through DSC in lines 231-232 of the paper? What is the full name of "Cp jump"? Additionally, "Xa" in line 233 should be "Xα".
- In lines 242-244 of the paper, what is the state of lactose during the grinding process in terms of its crystallinity, and will it have an impact on the quantification of Xam?
- In Figure 3, "Man." should be marked after the first appearance of Mannitol.
- In lines 244-246 of the paper, it is stated that the physical stability of the dispersion decreases with the increase of mannitol concentration. This conclusion should be supported by corresponding experiments or references.
Reviewer 3 Report
Comments and Suggestions for Authors
An interesting investigation, and well thought out explanation of the mechanisms!
There are some opportunities for grammatical improvements throughout. I found the abstract needing particular attention and have some recommendations below.
This sentence in the abstract does not read well. "Following a literature review, we monitored the milling kinetics of different pharmaceuticals compounds using ex-situ X-ray diffraction analysis in laboratory and in situ thanks to synchrotron facility, supplemented by DSC analyses". the 'Following a literature review' doesn't make sense. Was that literature review to select the compounds? that should be a separate sentence then. A literature review was conducted to select the model pharmaceutical compounds. Milling kinetics of the model compounds was monitored using X-ray diffraction analysis both ex-situ in the laboratory and in situ thanks to the synchrotron facility. X-Ray diffraction analysis was supplemented by DSC analyses.
"...mannitol and famotidine allow to conclude that..." should be allowed the conclusion, or allowed us to conclude. Also, why introduce mannitol here? it's not an active material? Is it participating in the crystallization?
I see later glycine is included as well. The title should maybe not be 'therapeutic' materials since the model compounds include both excipients and active ingredients.
These are long milling times. I see they were carried out at room temperature, but was the temperature of the material after milling measured? The milling chamber was not jacketed or cooled correct?
Was glycine the only compound studied in situ?
Also, the formatting of all of the figures varies significantly. Having formatting consistency across the figures would be helpful.
line 86 "...remain scare and their kinetics have been very little studied. It would therefore..." should be scarce
Reviewer 4 Report
Comments and Suggestions for Authors
The manuscript addresses the poorly understood mechanisms of milling-induced polymorphic transformations in pharmaceutical materials, with a systematic investigation of four model compounds, mannitol, famotidine, sulfamerazine and glycine. The use of both X-ray powder diffraction and synchrotron X-ray diffraction, supported by DSC, provides robust insights into the interplay between transient amorphization and recrystallization. The work advances the general mechanistic picture of such transformations, and the identification of a universal kinetic behavior is an important outcome with potential implications for formulation science.
That said, several aspects of the manuscript could be clarified to make the conclusions more robust and the message clearer to the reader.
1) The manuscript consistently capitalizes the names of chemical compounds (e.g., Mannitol, Glycine, Famotidine) even when not at the beginning of a sentence. In English scientific writing, compound names are normally written in lowercase unless they start a sentence or are proprietary names. Please check and standardize the capitalization throughout the text.
2) In the Introduction section, lines 113-115, the author’s states that “Among the four compounds chosen, two are enantiotropic (Glycine and Sulfamerazine) and two are monotropic (Famotidine, Mannitol).” Strictly speaking, compounds themselves cannot be described as monotropic or enantiotropic. These terms refer to the relationship between polymorphic forms of a given compound. This sentence should be rephrased.
3) In the Materials and Methods section, in 2.1, the milling ball used with the Retsch equipment is reported to be made of zirconium oxid. The milling jar is likely made of the same material, but this should be explicitly specified for clarity. In addition, the exact model of the Retsch mill should be indicated, as well as the milling conditions.
4) In the Results, in section 3.1, the authors state that “This indicates that part of the initial crystalline mannitol β has been coamorphized with the amorphous lactose while the other part has undergone a polymorphic transformation toward the form α”, based solely on the diffractogram presented in Figure 2c. However, at this stage, 12 h of milling, only weak α-mannitol peaks superimposed on a broad amorphous halo are visible. From this single diffractogram it is not possible to unambiguously conclude that two distinct processes, coamorphization and polymorphic transformation, occurred simultaneously. Such an interpretation requires the results presented in Figure 3. This sentence should be rephrased to avoid drawing mechanistic conclusions prematurely from Figure 2 alone.
5) It is not explain why the mannitol/lactose mixture was milled for 12 h and famotidine for 11 h. It would be helpful if the authors clarified the justification for these specific milling times.
6) In Figure 3, the amorphous fraction, Xam, is derived from DSC measurements of the Cp jump at Tg. It would be helpful if the authors could provide the corresponding DSC curves as supplementary material, at least for representative milling times, to support the presented quantification.
7) In Figure 4, it would be helpful to indicate that ‘”before milling’” corresponds to form B and “after 12 hours” corresponds to Form A.
8) In section 3.3, the sentence “The case of sulfamerazine is of particular interest, as the polymorphic forms involved in the transformations are enantiotropic, and such situations are only rarely documented in the literature”, is not very clear. It could be interpreted as suggesting that enantiotropic relationships between polymorphs are rare in general, which is not the case. What is less frequently documented is the study of milling induced transformations in systems with enantiotropic relationships. This sentence should be rephrased to avoid ambiguity.
9) Regarding Figure 5b, the authors state that “The thermogram recorded after 25 min of milling shows an exotherm ranging from 40 to 70°C which confirms this partial amorphization of the material”. However, if amorphization had occurred, one would expect to observe a glass transition in the DSC, visible as a step change in heat capacity, rather than a sharp exothermic peak. The observed exothermic signal is more likely due to cold crystallization following the devitrification of the partially amorphous material. Additionally, in the Introduction, the glass transition temperature of sulfamerazine is reported as 62 °C, yet the exothermic event in Figure 5b spans 40–70 °C. This discrepancy between the reported Tg and the temperature range of the exothermic event is not clearly addressed and may confuse the interpretation of the DSC data. The manuscript would benefit from clarifying whether the exothermic peak corresponds to cold crystallization and how this relates to the reported Tg of sulfamerazine.
10) Reference 13 also presents a study on the polymorphic transformation of sulfamerazine during milling at room temperature (Figure 6 in that reference). The authors should compare and discuss the results obtained in their study with those reported in Reference 13.
Round 2
Reviewer 1 Report
Comments and Suggestions for Authors
The authors thoroughly addressed all points. Only the section from line 600 to 613 about phase transitions, monotropy and enatiotropy still contains incorrect statements and requires revision before the manuscript can be accepted for publication.
I provide some explations, which which the authors should be able to quickly revise this section. I however recommed to the authors, that they also fimiliarize furhter with literature on enantiotropy, monotopy and Energy/Temperature diagrams fo polymorphic systems.
A monotropic relationship between a pair of polymorphs means, that one polymorph is more stable than the other ober the whole temperature range from absolute 0 to the melting point (of the lower melting polymorph). In such a system, a direct polymorphic transition in the solid state can only proceed from the less stable polymorph to the more stable polymorph. When during milling amorphous material is generated, which then recrystallizes, thermodynamics do not forbid the crystallization of amorphous to any of the two forms because the amorphous material is less table than both crystalline forms.
If we have an enantiotropic relationship between a pair of polymorphs, there is a thermodynamic transition point below the melting temperature. Below this thermodynamic transition point, one polymorph is more stable and above the thermodynamic transition point, the other form more stable.
This means, that below the thermodynamic transition point ony polymorph 1, which is less stable in this temperature region, can directly transform to the other in the solid state, while above the thermodynamic transition temperature, polymorph 2 is less stable and only this one can directly (i.e. solid-to-solid transition) transform to the other one. Again, if the transition proceeds via amorphous, this is not a direct transition of one polymorph to the other, and the amorphous material in principally free to crystallize in any of the two forms. If it follows Ostwad's "rule" of stages, the meatastable form will crystallize but it is also well prossible, that the more stable form crystallizes.
From my point of view, it would actually be very nice to have a statistic included on that, if not in this manuscript maybe in a later one.
Hope, this helps.
